# Inter- and Intramolecular RNA–RNA Interactions Modulate the Regulation of Translation Mediated by the 3′ UTR in West Nile Virus

**DOI:** 10.3390/ijms24065337

**Published:** 2023-03-10

**Authors:** Cristina Romero-López, Margarita Roda-Herreros, Beatriz Berzal-Herranz, Sara Esther Ramos-Lorente, Alfredo Berzal-Herranz

**Affiliations:** Instituto de Parasitología y Biomedicina “López Neyra”, Consejo Superior de Investigaciones Científicas (IPBLN-CSIC), 18016 Granada, Spain

**Keywords:** RNA–RNA interactions, West Nile virus, flavivirus translation, 3′ UTR, RNA virus, RNA dimerization

## Abstract

RNA viruses rely on genomic structural elements to accomplish the functions necessary to complete the viral cycle. These elements participate in a dynamic network of RNA–RNA interactions that determine the overall folding of the RNA genome and may be responsible for the fine regulation of viral replication and translation as well as the transition between them. The genomes of members of the genus *Flavivirus* are characterized by a complexly folded 3′ UTR with a number of RNA structural elements that are conserved across isolates of each species. The present work provides evidence of intra- and intermolecular RNA–RNA interactions involving RNA structural elements in the 3′ UTR of the West Nile virus genome. The intermolecular interactions can be visualized in vitro by the formation of molecular dimers involving the participation of at least the SLI and 3′DB elements. Certainly, the 3′ UTR of dengue virus, which lacks the SLI element, forms molecular dimers in lower quantities via a single interaction site, probably 3′DB. The functional analysis of sequence or deletion mutants revealed an inverse relationship between 3′ UTR dimerization and viral translation efficiency in cell cultures. A network of RNA–RNA interactions involving 3′ UTR structural elements might therefore exist, helping to regulate viral translation.

## 1. Introduction

RNA-mediated regulatory pathways have become the subject of great interest over the past two decades. The discovery of the importance of RNA–RNA interactions in the control of gene expression has allowed the deciphering of new information-coding systems across the entire phylogenetic spectrum, from viruses to humans. The small size of viral RNA genomes renders them important reservoirs of conserved structural elements that help in the maintenance of viral persistence without impeding adaptive fitness. These elements operate via the direct recruitment of cellular and viral protein factors and via the establishment of the inter- and intramolecular RNA–RNA interactions involved in complex and dynamic networks (the so-called RNA interactome). For example, many viral and cellular RNAs form dimers in order to enlarge their conformational space, thus improving their plasticity and functional abilities (for a review, see [1]). Further, RNA–RNA interactions play a role in the constitution of different ribonucleoprotein complexes that define the global structure of the genome and its function in the viral cycle, thus contributing to the formation of hierarchical regulatory systems. Understanding the functional role of these structures is crucial if we are to decipher the molecular basis of viral infections and develop efficient therapeutic strategies.

The genus *Flavivirus* is the largest in the family *Flaviviridae* and comprises a large group of viruses commonly referred to as flaviviruses. These are small, enveloped, positive single-stranded RNA viruses. Several are human pathogens, including dengue (DENV), Zika (ZIKV), yellow fever (YFV) and West Nile virus (WNV), all of which belong to the mosquito-borne flavivirus (MBFV) group [2,3,4]. The information required by MBFVs for the successful completion of their infectious cycle is encoded in the approximately 11,000 nt that constitute their genome. It bears a single open reading frame (ORF) coding for a polyprotein that is processed to yield three structural and seven non-structural proteins (Figure 1). The ORF is flanked by untranslated regions (5′ and 3′ UTRs) that vary in length and structure among flaviviruses (Figure 1). Each virus species has complexly folded RNA domains that are strongly conserved across isolates.

Evidence exists that specific RNA elements in the 3′ UTR help regulate essential viral processes, including translation, replication and infectivity [5,6,7], as well as the transitions between them. During early infection, the viral genome is used as mRNA to initiate viral protein synthesis at the capped 5′ UTR. The efficiency of translation depends on structural elements located in the 3′ UTR [7,8,9], although the molecular mechanism underlying this remains unknown. Among others, these elements include SL-III, the so-called 5′ dumbbell (5′DB) and the CS1 sequence. The last two participate in the formation of functionally important tertiary structures, such as the pseudoknots PK3 and PK4 (Figure 1) [6]. The initiation of replication also requires the intact 3′ UTR to generate a cyclized genome dependent on the establishment of direct RNA–RNA interactions between the 5′ and the 3′ ends [10,11,12], which are sensed by the pseudoknots of the 3 ’UTR [5,6]. Such a dual regulatory role for the 3′ UTR may mean that it manages the switches between the different stages of the viral cycle.

The hepatitis C virus (HCV), which also belongs to the family *Flaviviridae*, has been shown to form genomic dimeric units mediated by a palindromic sequence motif (dimer linkage sequence (DLS)) [13] located in the 3′ UTR of HCV RNA. These dimers function as preferential templates for the viral RNA polymerase [14]. We have shown that genomic HCV variants with reduced dimerization capacity also show reduced translation efficiency [15]. Thus, genomic dimeric units in HCV are necessary for the control of essential processes during the infectious cycle [14,16].

Here, it was hypothesized that, since HCV and flaviviruses show similar genetic organization and an essential information storage system based on discrete structural and functional units, they may also share common molecular mechanisms for the regulation of viral processes. Although no DLS has been identified in the flaviviral genome, RNA–RNA assembly may occur by means other than Watson–Crick and non-Watson–Crick base pairing. For example, base stacking and helix co-stacking are commonly used to establish RNA–RNA contacts in crowded RNA environments [17]. Thus, unless steric impediments exist, intermolecular RNA–RNA interactions may occur in any biological context involving high RNA concentrations. Such concentrations may arise during active viral replication, when large viral RNAs with complex structural motifs are synthesized. These complexes help ribonucleoprotein assemblages generate stress granules enriched in translation initiation factors and 40S ribosomal subunits, as well as P-bodies containing proteins involved in RNA turnover (for a review, see [18]). In summary, flaviviral RNA dimerization might help regulate multiple stages of the infectious cycle, as well as the transitions between them.

This work provides evidence of the in vitro formation of molecular dimers in WNV and DENV, a process reliant on specific structural elements of the 3′ UTR RNA, and proposes the existence of a complex network of intra- and intermolecular contacts that needs to be balanced in order to achieve the control of translation. Together, the data presented support a plausible functional role for the phenomenon of genomic RNA dimerization in flavivirus infectious cycles.

## 2. Results

### 2.1. The 3′ UTR of WNV Dimerizes In Vitro

The 3′ UTR of HCV is known to dimerize in vitro [19,20]. Here, it was hypothesized that this phenomenon might be a general feature of members of *Flaviviridae* and that it might operate as a regulatory mechanism.

To test this hypothesis, increasing concentrations of an RNA construct encoding the full 3′ UTR of WNV (named 3′WNV) were incubated in the presence of Mg^2+^. Different conformations were resolved by agarose gel electrophoresis under non-denaturing conditions. Up to four different conformers were detected at the highest tested concentrations of 3′WNV (Figure 2a), reflecting the existence of concentration-dependent conformational transitions. These results showed a dimerization efficiency close to 80% (Figure 2a). Data were fitted to a non-linear equation (see Figure 2a).

Inter- and intramolecular RNA interactions are stabilized by the presence of Mg^2+^ and, to a lesser extent, by monovalent cations such as Na^+^ [21]. The requirement of these ions for the dimerization of 3′ UTR WNV RNA was therefore analyzed. Dimerization assays were conducted in the absence or presence of increasing concentrations of either Mg^2+^ or Na^+^ (Figure 2b,c). A positive dependence on divalent cations for dimer formation was observed for low concentrations of Mg^2+^, while at higher Mg^2+^ concentrations, the formation of dimers was impaired (Figure 2b). This suggests that the stabilization of certain conformations in the monomeric form at high Mg^2+^ concentrations may impede their progression toward dimeric forms. In contrast, an increase in Na^+^ had a positive effect on dimer formation (Figure 2c) at all concentrations tested (see Materials and Methods section), likely by stabilizing conformations susceptible to dimerization. Taken together, these results show that the 3′ UTR of the WNV genome can form dimers in vitro in the absence of protein factors.

### 2.2. Structural Elements Essential for the Dimerization of the 3′ UTR, as Revealed by HMX

The 2′-hydroxyl molecular interference (HMX) technique [22] was used to obtain information on the nucleotides that are necessary for dimer formation to occur, either because they are directly involved in dimerization or because they participate in the formation of the competent structure for this interaction. A comparison of nucleotide reactivity between the monomers and dimers will provide information about the nucleotides involved in the formation of dimers.

Briefly, the RNA construct encompassing the whole 3′WNV was modified with NMIA under denaturing conditions. RNA molecules were then renatured and subjected to dimerization assays under optimal experimental conditions. The monomeric and dimeric populations were then separated by native agarose gel electrophoresis. Modified positions were detected via stoppages in the reverse transcription reaction. HMX scores were calculated from the differential reactivity profiles between the monomeric and dimeric conformers (for more details, see Materials and Methods).

HMX analysis showed the need for precise nucleotide conformations in essential structural elements of the 3′ UTR for the dimeric complex to form (Figure 3a,c). The extensive changes in NMIA reactivity mapped in domain II and the SL-II-RCS3 elements are noteworthy (Figure 3b,c). These were mainly reductions in the dimer conformer, suggesting these elements need to be structurally stable for dimer formation to occur. In addition, these elements might directly participate in dimer formation. Minor changes in NMIA reactivity were also detected throughout the entire 3′ UTR, suggesting the dimerization process to be affected by the establishment of distant and different RNA–RNA interactions in this region, indicating that the interactome defines the structure of the full-length RNA.

RNAStructure software [23] was then used to create a theoretical model of the 2D structure of the dimeric conformer (Figure 3c). The latter figure shows that, according to the NMIA reactivity data, the 3′DB element is a potential dimerization site in the TL2 loop. Two sequence motifs in SL-I (positions A10,435-C10,441 and G10,472-U10,478) were also predicted to interact with their corresponding complementary sequences downstream of the 3′DB element (positions G10,929-U10,935 and A10,912-C10,919, respectively). These interactions might be established in an intra- and intermolecular manner, likely promoting an equilibrium between the monomeric and dimeric conformers.

Taken together, these results support the existence of new RNA–RNA interactions that occur in the 3′ UTR of the WNV genome, and that these might be critical for preserving the functional interactome at the 3′ end of viral RNA.

### 2.3. The SL-I and 3′DB Elements Are Essential for Efficient 3′ UTR Dimer Formation

Structural determinants for dimer formation have been shown to map to different elements of the 3*′* UTR, such as 3*′*DB, as detected by HMX. The participation of SL-I in the formation of intra- and intermolecular interactions is also suggested from the secondary structure prediction. To confirm the role of these elements in dimer formation, a collection of mutant constructs was generated by the sequential deletion of the autonomously folded structural elements within the 3*′* UTR. These deletions were generated in both the 3*′* end (named Mut 1-5_3*′*; Figure 4a) and 5*′* end (the so-called Mut 1-5_5*′*; Figure 4b). Dimerization assays were performed as described above, and the reaction products were resolved by native agarose gel electrophoresis.

Sequential deletions at the 3*′* end showed the removal of the 3*′*DB element to completely abolish the dimerization ability of the 3*′* UTR (see Mut 3_3*′*, Figure 4a). Eliminating SL-I in Mut 1_5*′* produced the same effect (Figure 4b). These results confirm the essential role of 3*′*DB and SL-I in dimerization and suggest them to be the main anchoring sites in intermolecular interactions.

Intriguingly, the Mut 5_3*′* and Mut 3_5*′* constructs partially recovered the ability to dimerize (Figure 4), although never matching that of the wt 3*′* UTR. Mut 3_5*′* retains the double dumbbell flanked by the highly stable elements PK2 and 3*′*SL (Figure 4b). Such a stable structural environment might provide favorable conditions for dimer formation at 3*′*DB. Mut 5_3*′*, which barely dimerizes, preserves SL-I, followed by the PK1 and SL-III elements, which were shown by HMX to be involved in dimer formation (Figure 4a). It is likely that, in this structural context, SL-I operates as a minor dimerization anchoring site at which dimer formation would be mediated by pseudopalindromic, asymmetric sequence motifs. In addition, a fraction of molecules of Mut5_3*′* showed retarded migration, suggesting the existence of aggregates or populations with suboptimal folding.

### 2.4. Specific Sequences in SL-I and 3′DB Mediate 3′ UTR Dimerization

The HMX structural data, theoretical models determined using RNAStructure software, and functional dimerization assays all showed the importance of SL-I and 3*′*DB in dimer formation. These findings were used to design a new set of constructs containing mutations at the precise positions of SL-I, 3*′*DB and the 3*′* sequence of double-dumbbell domain II. All of the mutants were generated by flipping the sequence motifs predicted by RNAStructure to be involved in the establishment of intra- and intermolecular interactions (Figure 5). None of the introduced mutations altered the theoretical secondary structures of the structural elements, as predicted by RNAfold [24].

The importance of contact 10,436-10,929 (Figure 3c) in dimer formation was tested by constructing the following mutants: the construct Mut_Flip_SLIa, which mutates positions 10436-10441 to interrupt the potential interaction with the CS1 sequence (positions 10,929-10,935); Mut_Flip_CS1, which flips the sequence in position 10,929-10,935; and the construct Mut_Flip_SLIa+CS1, which contains a double mutation that restores the theoretical interaction. In vitro dimerization assays were conducted as described above. A reduction in dimerization efficiency was seen for Mut_Flip_SLIa, highlighting the key role of the sequence motif at position 10,436 (Figure 3a and Figure 5a). Interestingly, no changes in dimer formation were detected for Mut_Flip_CS1 compared to the wt 3*′* UTR, while the compensatory mutation carried by Mut_Flip_SLIa+CS1 partially restored the dimerization capacity (Figure 5a). These results indicate that the formation of dimers mediated by SL-I does not involve different sequences in each interacting molecule, but rather a single motif located at the 5*′* apical portion of SL-I.

Sequence 10436-10441 in the apical portion of the stem of SL-I is pseudopalindromic. To examine the potential role of the apical stem and closing loop in dimer formation, two constructs were generated (Figure 5b): Mut-SLI_GNRA, which bears a GNRA substitution in the apical loop of SL-I, and Mut-delSt1, with a deletion involving positions 10,432-10,441+10,553-10,662 in the apical portion of the stem of SL-I (Figure 3c and Figure 5b). Dimer formation in these constructs was completely abrogated, confirming the need for the SL-I apical fragment. More importantly, these results also show that the apical loop of SL-I is an important partner in 3*′* UTR dimerization, suggesting a kissing-loop interaction to be involved.

The constructs Mut-Flip_SLIb and Mut-Flip_10912, which carry inversions of the sequences in positions 10,472 and 10,912, respectively, were generated to validate contact 10,472-10,912 (Figure 3c and Figure 5c). No variation in dimerization capacity was seen for these RNAs compared to 3*′*WNV RNA, suggesting that this contact is not required for the generation of a dimeric conformer (Figure 5c).

Finally, Mut_d3DB, which lacks the 3*′*DB element, was constructed (Figure 5d). This molecule was associated with a significant reduction in dimerization capacity with respect to 3*′*WNV, confirming the 3*′*DB element to be a critical interaction site in the formation of dimeric conformers.

### 2.5. Dimerization Is also Observed for the (−) Strand of the WNV Genome

During viral replication, emerging genomic RNA molecules might use the dimerization process as a means of controlling the late steps of progression through the flavivirus cycle. In this context, the role of the (−) strand has been suggested to be critical for enhancing the synthesis of the (+) strand [25]. The ability of the (−) strand to form dimeric complexes was therefore tested. For this, an RNA construct corresponding to the complementary sequence of the 3*′* UTR (the so-called (−) 3*′*UTR) was synthesized and subjected to dimerization assays using increasing RNA concentrations. Data were fitted to a non-linear curve, as described for the (+) strand. The results showed a low yield of dimer complex formation (~50%) (Figure 6).

### 2.6. New Roles for the SL-I and 3′DB Elements in the Control of WNV Translation

As has been mentioned, evidence exists that specific RNA elements in the 3*′* UTR regulate viral translation [7]. In an attempt to correlate the interference in dimerization with a potential effect on 5*′*UTR-dependent protein synthesis mediated by the 3*′* UTR, a new set of constructs was designed containing the 5*′* UTR of WNV fused to the reporter gene encoding the FLuc protein (Figure 7a). Different 3*′* UTR “Flip” variants that promoted variations in dimer formation (Mut_Flip_SLIa, Mut_Flip_CS1, Mut_Flip_CS1 and Mut_d3DB) were attached to the 3*′* end of the FLuc coding sequence to generate WNV-Mut_X constructs (Figure 7a). All of these RNA molecules were 5*′*-capped and used to transfect Vero or C6/36 cells along with RNA coding for the reporter protein RLuc, the translation of which is also cap-dependent. This allowed for the reliable comparison of transfection efficiency (see Materials and Methods).

Interestingly, mutations in SL-I or CS1 induced a significant increase in FLuc synthesis in both cell types, an effect that was even stronger for WNV-Mut_CS1 in mosquito cells (Figure 7b). More importantly, restoring the interaction SL-I+CS1 did not restore FLuc activity to that of the wt WNV construct (Figure 7b), suggesting that the sequence motifs, and not the interaction itself, are the main agents responsible for this enhancing effect. Again, in C6/36 cells, a stronger increase in translation was detected for the WNV-Mut_SLIa+CS1 construct (Figure 7b), suggesting the existence of cellular factors that influence its effect on protein synthesis.

These studies also revealed a new role for the 3*′*DB element in WNV translation. Its deletion reduced WNV translation efficiency by up to 50% (Figure 7b) in both cell types, revealing it to be a translation enhancer.

In summary, mutations in SL-I and CS1 promote opposite effects on dimerization and viral translation, suggesting that the interaction giving rise to the dimer serves different regulatory functions in flaviviruses.

### 2.7. Genomic Dimerization in Flavivirus: DENV and YFV

To determine whether RNA dimerization mediated by the 3*′* UTR might be a general feature of MBFVs, constructs bearing the 3*′* UTRs of two WNV-related flaviviruses—DENV and YFV—were examined for their ability to form dimers (Figure 8a) under similar conditions to those set in the WNV dimerization assays. While the 3*′* UTR of DENV was shown to form dimeric complexes, the closely related YFV did not, indicating that dimerization mediated by the 3*′* UTR is not generalized across all flaviviruses.

In further work, increasing concentrations of 3*′*DENV RNA were incubated under dimerization conditions. The proportion of dimers was quantified, and the data were fitted to a non-linear equation, as described for 3*′*WNV RNA (Figure 8b). Interestingly, 3*′*DENV RNA formed dimers less efficiently than did 3*′*WNV. These observations might be related to the fact that the 3*′* UTR of DENV carries the 3*′*DB element, but the SL-I element is absent, reinforcing the idea of independent dimerization sites proposed for WNV.

## 3. Discussion

RNA assembly through the establishment of complex networks of interactions is central to the regulation of numerous biological processes [18,26,27]. The dimerization of viral RNA genomes, initially demonstrated in retroviruses, is perhaps the most studied intermolecular interaction of all [28]. Such findings opened the door to investigating dimerization phenomena in other RNA viruses. It is now known that SARS-CoV-1 undergoes an intermolecular interaction via a palindromic sequence located in a complex three-stem pseudoknot structure, leading to the stimulation of a functional frameshifting event and the regulation of the relative abundance of viral RNA transcripts [29]. The HCV genome can form dimers too, a process mediated by a palindromic sequence (DLS) located at the 3*′* end of the viral genome and related to efficient translation and replication [14,15]. The present work provides the first evidence that RNA dimerization also occurs in MBFVs such as WNV and DENV, mediated by complementary sequences in the 3*′* UTR of the viral genome. Importantly, the sequences involved in this dimerization are also key elements for the translational regulation exerted by the 3*′* UTR, suggesting a regulatory role for dimer formation in flaviviruses.

The present structural and mutational analyses identified the sequence motifs involved in establishing intra- and intermolecular RNA interactions. These are located in the apical portion of SL-I and the 5*′* portion of the 3*′*DB element, including TL2, which interacts with the 5*′* portion of CS1 to generate PK4. The deletion of the SL-I element prevents dimerization, while the suppression of 3*′*DB significantly reduces the dimerization yield. This indicates that both elements are important for this process, although to different extents (Figure 5a,c). It is possible that the balance of the interactions involving SL-I and CS1 could be displaced from intramolecular to intermolecular contacts at high viral RNA concentrations, such as those inducing molecular crowding in replication complexes. These interactions, though thermodynamically plausible, would not be stable enough to support the formation of dimeric conformers, but they would induce the release of 3*′*DB from PK4, permitting more dynamic folding. This dynamic folding would favor a more stable interaction involving the 3*′*DB element to generate dimers. Thus, dimerization in WNV might involve complementary strand swapping and kissing-loop interactions at the apical portion of SL-I, as well as kissing-loop interactions at TL2 (Figure 5). Further dimerization analyses including point mutations in the TL2 sequence will be carried out in order to precisely identify the nucleotides required for this interaction. Finally, it is likely that multiple structural elements within the 3*′* UTR operate in a coordinated manner to control the exposure of the dimerization sequences.

The locations of SL-I and 3*′*DB, and their relationships with other elements of the 3*′* UTR, suggest the existence of a complex network of intra- and intermolecular contacts that structurally tunes the whole 3*′* UTR, thus determining its functions. The fact that many of the sequence motifs involved in this network overlap suggests that the preservation of an equilibrium between these interactions is required to control the different steps of the viral cycle. In DENV, it has been reported that the pseudoknot including the 3*′*DB element and the CS sequence may operate as a sensor to regulate direct 5*′*–3*′* interactions in the viral genome that are essential for the initiation of replication [5,6]. Importantly, in the present work, the construct WNV-Mut_CS1 doubled and tripled the translation efficiency of WNV in Vero and mosquito cells, respectively. This molecule carries a reverse CS1 sequence that impedes the intramolecular interaction with SL-I and the formation of the pseudoknot PK4 with the 3*′*DB element. It may be that, during early translation, the formation of PK4 acts as a translation repressor, most likely by impeding the acquisition of proficient translational conformers at the 3*′* UTR. This would be consistent with a previous report showing CS1 to be a key motif for translational repression in DENV [8]. As proposed by other authors [30], switching from the sequestration of CS1 to its release, mediated by the establishment of RNA–RNA contacts, could act as a regulatory mechanism for controlling transitions between different steps of the viral cycle.

The results show that sequences that affect dimerization also affect translation, although in different ways. For example, the requirement of SL-I in WNV for efficient dimerization, as well as its relationship with viral translation, is intriguing. We demonstrate that modifications in SL-I that impede efficient dimerization also lead to an increase of ~2-fold in translation (Figure 5 and Figure 7). In the closely related ZIKV, SL-I recruits the protein Msi, which favors the inhibition of translation and the accumulation of viral genomes [31]. This is in good agreement with the results presented in Figure 7b, which show that, in WNV, SL-I is a translation repressor. Interestingly, the SL-I of ZIKV can interact with proteins related to stress granules, such as Caprin-1, G3BP1, G3BP2 and USP10, to assess the success of infection [31]. Stress granules are dynamic structures that form under different stress conditions and dissolve when homeostasis is restored. Flaviviral infection induces the formation and determines the composition of stress granules in a unique way. These granules aggregate when viral RNAs accumulate at replication centers and contain pre-stalled 43S and 48S translation complexes [32,33,34,35]. In this context, dimer formation in WNV would likely regulate viral protein synthesis and viral RNA stability. Interestingly, contrary to what was observed for SL-I, the deletion of 3*′*DB leads to a deficit in both dimerization and translation. These results support that the control of WNV translation mediated by the 3*′* UTR is a multifactorial phenomenon that depends on the establishment of a proper balance between different elements, as previously described [7].

It is noteworthy that the 3*′* UTR of DENV also formed dimeric complexes, although to a lesser extent than did that of WNV (Figure 8a). This can be explained by the fact that the 3*′* UTR of DENV does not bear the SL-I element; therefore, dimerization is mediated by the 3*′*DB element only. While the dimeric form may be just as stable, the dimerization rate would be reduced. Other authors have reported the existence of molecular dimers mediated by the 3*′*DB element of a closely related MBFV (these dimers were serendipitously crystallized and subjected to X-ray analysis [36]). From a functional point of view, it has been observed that the 3*′* UTR of DENV operates as a potent translation enhancer of WNV 5*′* UTR-dependent protein synthesis [7]. Taken together, these results support the assumption that dimerization and translation are intimately connected events in WNV and DENV. In contrast, YFV lacks the key dimerization sites detected in the 3*′* UTR of WNV (SL-I and 3*′*DB) (Figure 1), leading to incompetent dimerization. Nevertheless, genomic structural rearrangement strategies might be used by YFV and related MBFVs to achieve dimer conformers using RNA elements outside of the 3*′* UTR.

Taken together, the present results indicate the existence of genomic RNA dimerization mediated by the 3*′* UTR of certain MBFVs. In WNV, this occurs at two different anchoring sites with different affinities and could overlap with the establishment of previously unreported, theoretical intramolecular interactions predicted with RNAStructure (Figure 3c). A complex network of contacts involving the 3*′* UTR of WNV is proposed, which must be carefully balanced if the regulation of viral translation and replication is to be achieved.

## 4. Materials and Methods

### 4.1. DNA Templates and RNA Synthesis

The plasmid pUC3*′*WNV was constructed as follows. Firstly, DNA encoding the 3*′* UTR of the WNV genome (3*′*WNV) was generated by PCR using the primers T7p3*′*WNV and as3*′*WNV-BamHI from the modified NY99-flamingo382–99 strain [37], a kind gift from Dr. J.C Saiz (INIA, Madrid, Spain). These primers included restriction sites for the EcoRI and BamHI endonucleases, respectively. In addition, the T7p3*′*WNV oligonucleotide contained the promoter sequence of T7 RNA polymerase after the EcoRI site. The resulting amplification product was cloned into EcoRI-BamHI sites of the pUC19 vector to generate pUC3*′*WNV. The latter was digested using the endonuclease BamHI to generate the template for subsequent 3*′*WNV RNA synthesis.

The DNA fragments coding for the 3*′* UTRs of DENV (NC_001474.2) and YFV (KF769016.1), fused to the T7 promoter sequence, were purchased from IDT (Coralville, IA, USA). These constructs were introduced into the plasmid pUC19, which was first linearized by BamHI using the In-Fusion*^®^* HD Cloning Kit (Takara, Kusatsu, Japan), to generate pUC3*′*DENV and pUC3*′*YFV. These plasmids were linearized with the enzyme SalI to produce templates for transcription.

DNAs encoding sequential deletions in the 3*′*WNV were generated by PCR amplification from the pUC3*′*WNV vector. For 3*′* deletions, the T7p3*′*WNV oligonucleotide was used as a 5*′* primer for all constructs, while the 3*′* primers were as follows: asWNV-10950 to generate DNA T7pMut1_3*′*, asWNV-10904 for DNA T7pMut2_3*′*, asWNV-10830 for DNA T7pMut3_3*′*, asWNV-10740 for DNA T7pMut4_3*′*, and asWNV-10660 for DNA T7pMut5_3*′*. To generate different DNA constructs bearing 5*′* deletions, the 3*′* primer asWNV-BamH1 was always used, while the 5*′* oligonucleotides used were as follows: T7pWNV-10500 to yield DNA T7pMut1_5*′*, T7pWNV-10586 for DNA T7pMut2_5*′*, T7pWNV-10659 for DNA T7pMut3_5*′*, T7pWNV-10743 for DNA T7pMut4_5*′*, and T7pWNV-10831 for DNA T7pMut1_5*′*. The resulting amplicons were used as DNA templates for RNA synthesis (see below).

The DNA coding for (-) 3*′*WNV was generated by PCR using the oligonucleotides rc3*′* WNV and T7 GGG- asrc 3*′*WNV, with pUC3*′*WNV as a DNA template. The amplification product carried the T7 promoter sequence fused to the 5*′* end of the first 625 nt of the negative strand of the WNV genome, i.e., the complementary sequence of the 3*′* UTR. This DNA was used for subsequent in vitro transcription (see below).

All of the above-mentioned DNAs were used as templates for the synthesis of their respective constructs by in vitro transcription using T7 RNA polymerase ver.2.0 (Takara). RNAs were purified, and their quality was analyzed as previously described [38].

DNA plasmids carrying “Flip” mutations derived from the parental pUC3*′*WNV (pUC3*′*WNV-Flip series) or pGLWNV (pGLWNV-Flip series) [7] vectors were obtained by site-directed mutagenesis using the Phusion™ Site-Directed Mutagenesis Kit (Thermo Fisher Scientific, Waltham, MA, USA) with the appropriate primer pairs and following the manufacturer’s instructions. DNA molecules corresponding to the “pGLWNV-Flip” series carry the FLuc coding sequence—but lack its AUG start codon—flanked by the 5*′* UTR of the NY99-flamingo382–99 WNV strain and the 3*′* UTR of WNV with the indicated Flip mutations. These constructs were designed to allow fluc translation from the AUG of the 5*′* UTR of WNV.

pUC3*′*WNV was used as a template for site-directed mutagenesis to generate two constructs: Mut-SLI_GNRA, which substitutes the apical loop of SL-I for a stable GNRA tetraloop, and Mut_delSt1, which does not have the apical portion of the stem of SL-I. Mutations were introduced using the appropriate primers and employing the Phusion™ Site-Directed Mutagenesis Kit (Thermo Fisher Scientific).

RNA constructs derived from pUC3*′*WNV were synthesized by in vitro transcription from the corresponding BamH1-linearized plasmids, as described above. The RNAs derived from the pGLWNV-Flip series bearing the 5*′* cap were obtained using the HighYield T7 RNA Synthesis Kit (Jena Bioscience, Jena, Germany) following the manufacturer’s instructions [39]. The synthesis of cap-RLuc RNA was performed as previously described [39]. RNAs were purified and their quality was assessed as described elsewhere [7].

### 4.2. Dimerization Assays

RNA dimer formation was performed essentially as previously described [16]. Briefly, RNAs were denatured in water by heating at 95 °C for 2 min and then snap cooling on ice for 15 min. Increasing concentrations of RNA molecules were incubated in SHAPE buffer (20 mM HEPES pH 8.0, 100 mM NaCl, 1 mM MgCl_2_) [40] for 30 min at 37 °C. The reaction products were then resolved by native agarose gel electrophoresis (1.2–1.5% *w*/*v*) in TBM buffer (45 mM Tris-HCl pH 8.3, 43 mM boric acid, 0.1 mM MgCl_2_) [41] at 4 °C and 6 V/cm for 3 h. The RNA conformers were visualized by staining with RedSafe™ reagent (Labotaq, Seville, Spain) following the manufacturer’s instructions and further detected under UV trans-illumination. Quantification was performed using Image Lab™ software (Bio-Rad, Hercules, CA, USA). Data were fitted to the equation y = ((B_max1_)/(K_d1_ + x)) + ((B_max2_)/(K_d2_ + x)), where y is the percentage of dimeric conformers, B_maxn_ is the maximum yield of the reaction, x is the concentration of 3*′*WNV RNA, and K_dn_ the concentration of RNA required to achieve the maximum dimer percentage.

Ionic requirements were studied via dimerization assays performed as above but with appropriate reaction buffers. Thus, to test Mg^2+^ requirements, the reaction buffer included EDTA (20 mM HEPES pH 8.0, 100 mM NaCl, 1 mM EDTA pH 8.2) and increasing concentrations of Mg^2+^ (0, 0.1, 0.2, 0.5, 1, 2 and 5 mM), while for monitoring Na^+^ requirements, monovalent cations were added to the reaction buffer (20 mM HEPES pH 8.0, 1 mM MgCl_2_) in the range 2*–*200 mM (0, 2, 10, 50 100 and 200 mM).

### 4.3. HMX (2′ Hydroxyl Molecular Interference) Assays

HMX analyses were performed as previously described with minor modifications [16]. Briefly, 25 pmol of each RNA construct was denatured at 95 °C for 2 min in 18 μL of 100 mM HEPES pH 8.0 and subsequently cooled at 4 °C for 15 min. RNA was then modified with 30 mM N-methylisatoic anhydride reagent (NMIA) at 95 °C for 2 min. Samples were then cooled on ice for 5 min. This process was repeated three times. RNA molecules were precipitated with absolute ethanol and then washed once with 80% ethanol. Total RNA was monitored by UV spectrophotometry (A_260_). Modified RNAs were subjected to dimerization assays as described above. Complexes were also resolved as described above. The monomeric and dimeric conformers were gel-purified and used as templates in a primer extension reaction with the appropriate fluorescently labeled oligonucleotide to detect NMIA modifications by capillary electrophoresis, as previously described [16]. Electropherograms were analyzed using QuShape software, and relative reactivity values at each nucleotide position were calculated as previously described [42]. HMX scores were calculated as previously reported [16,22].

### 4.4. Cell Culture

Vero cells, derived from African green monkey kidney cells, were cultured as previously described [7]. Briefly, minimum essential medium (MEM) was supplemented with 5% heat-inactivated fetal bovine serum (FBS; Gibco*^®^* by LifeTechnologies™, Invitrogen, Waltham, MA, USA), 2 mM L-Glutamine (Sigma, St. Louis, MO, USA) and 1 mM sodium pyruvate (Sigma). Cells were grown at 37 °C in a 5% CO_2_ atmosphere.

Aedes albopictus larva C6/36 cells were cultured in M3 insect medium supplemented with 10% heat-inactivated FBS, 2 mM L-glutamine and 0.1 mM non-essential amino acids, as previously described [7], and maintained at 28 °C in a 5% CO_2_ atmosphere.

### 4.5. Cell Transfection and Luciferase Assays

Cells were transfected essentially as previously described [7,39]. Briefly, 60 × 10^3^ cells were seeded into 24-well plates 24 h before transfection in order to reach ~80% confluence. Then, 1.5 µg of the RNA under study and 0.3 µg of cap-RLuc RNA were mixed with the transfection reagent (TransFectin™; Bio-Rad) in Opti-Mem*^®^* (Gibco*^®^* by LifeTechnologies™, Invitrogen) according to the manufacturer’s instructions. This mix was added to cells followed by incubation for 4 h in order to achieve maximum translational efficiency. The cells were then lysed and subjected to firefly and Renilla luciferase assays using the Dual-Luciferase Reporter Assay Kit (Vazyme Biotech, Nanjing, China). Relative translation efficiency was calculated as the FLuc/RLuc ratio compared to that obtained for wt WNV RNA.

## Figures and Tables

**Figure 1 ijms-24-05337-f001:**
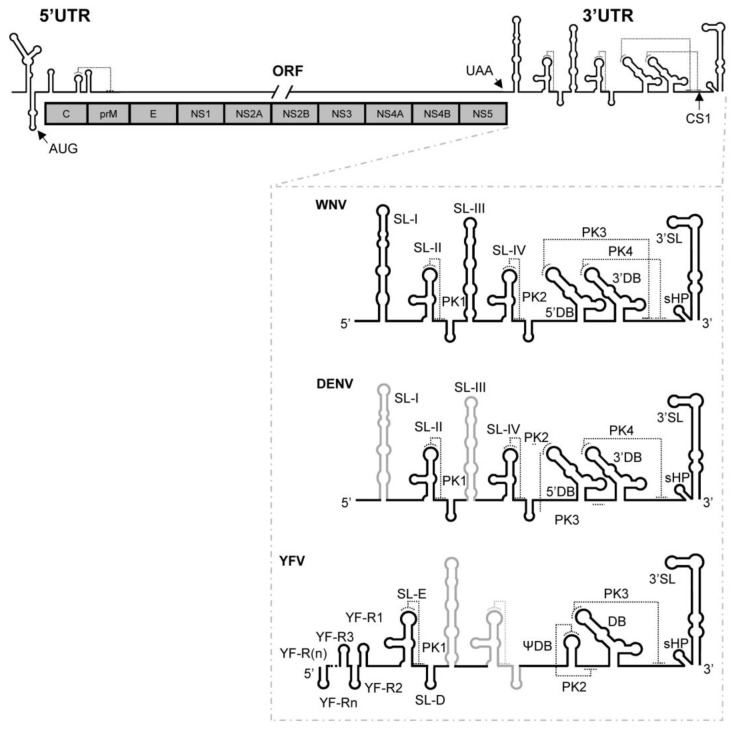
Structural elements described in the 3′ UTR of representative flaviviruses. General scheme of the secondary structure of the flavivirus RNA genome. Below are shown the secondary structure models of the genomic 3′ UTRs of the flaviviruses WNV, DENV and YFV used in this work. The identified highly conserved secondary structural elements, tertiary interactions and the CS1 sequence motif are indicated. Differences in the structural organization of the 3′ UTR of DENV and YFV compared to that described for WNV are denoted as thick black lines on the secondary structure model of WNV (thick gray lines). Secondary and tertiary structural elements are named. Thin dotted lines indicate pseudoknot interactions. In YFV, thick dotted lines represent extra RYF elements in different members of the clade.

**Figure 2 ijms-24-05337-f002:**
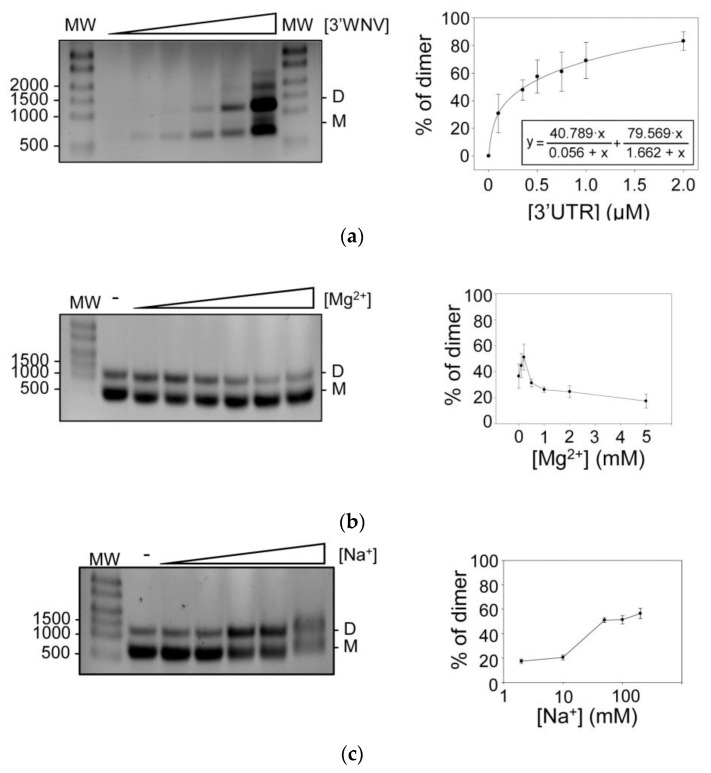
Dimerization of the 3′ UTR of the WNV RNA genome in a Mg^2+^- and Na^+^-dependent manner. (**a**) Left panel: the dimerization efficiency of 3′WNV RNA was assayed at different concentrations. Different conformers were resolved by native agarose gel electrophoresis. A representative image of these assays is shown. Relative dimer formation was quantified (right panel). Data represent the mean of at least three independent experiments ± standard deviation. (**b**) and (**c**) Mg^2+^ and Na^+^ dependence for dimer formation. Dimerization assays were performed in the presence of increasing concentrations of Mg^2+^ (**b**) or Na^+^ (**c**) ions. Conformers were resolved as noted above (left panels). Relative dimerization was quantified in order to establish the optimal ionic conditions. Data represent the mean of at least three independent experiments ± standard deviation (**b**). M, monomer; D, dimer.

**Figure 3 ijms-24-05337-f003:**
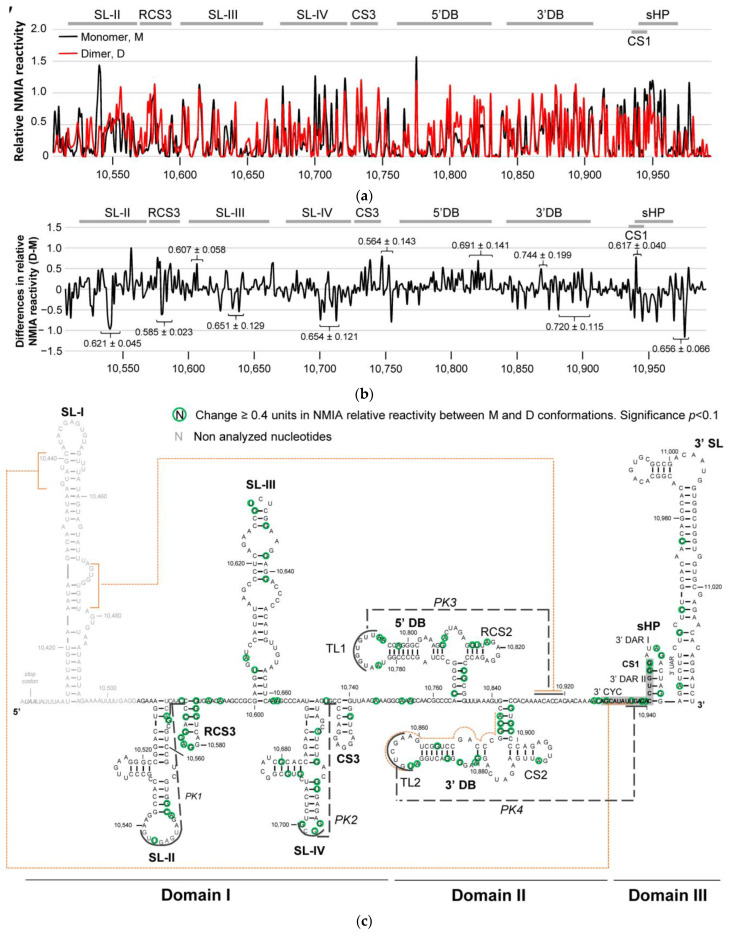
Identification of the structural determinants for dimer formation in the 3′ UTR of WNV. Structural requirements at single nucleotide positions were detected by 2′-hydroxyl molecular interference (HMX). (**a**) The graph shows the NMIA reactivity profile under denaturing conditions for each of the isolated conformers of the 3′WNV molecule. The monomeric and dimeric forms were partitioned by native agarose gel electrophoresis. Modified positions were identified as stops by reverse transcription. Data are the mean of four independent experiments ± standard deviation. M, monomer, solid black line; D, dimer, solid red line. (**b**) Differential NMIA reactivity profile for the dimer vs. monomer forms. HMX scores indicated at precise positions were calculated from the reactivity profiles of the monomeric and dimeric conformers, as previously described [22]. (**c**) Sequence and secondary structure of 3′WNV RNA indicating the main HMX results. Green circles indicate a significant (*p* < 0.1) increase or decrease of ≥0.4 NMIA reactivity units. Reactivity data were used to predict a theoretical structural model, including intra- and/or intermolecular interactions (depicted as orange dashed lines), with the software RNAStructure. The CS1 sequence is shadowed.

**Figure 4 ijms-24-05337-f004:**
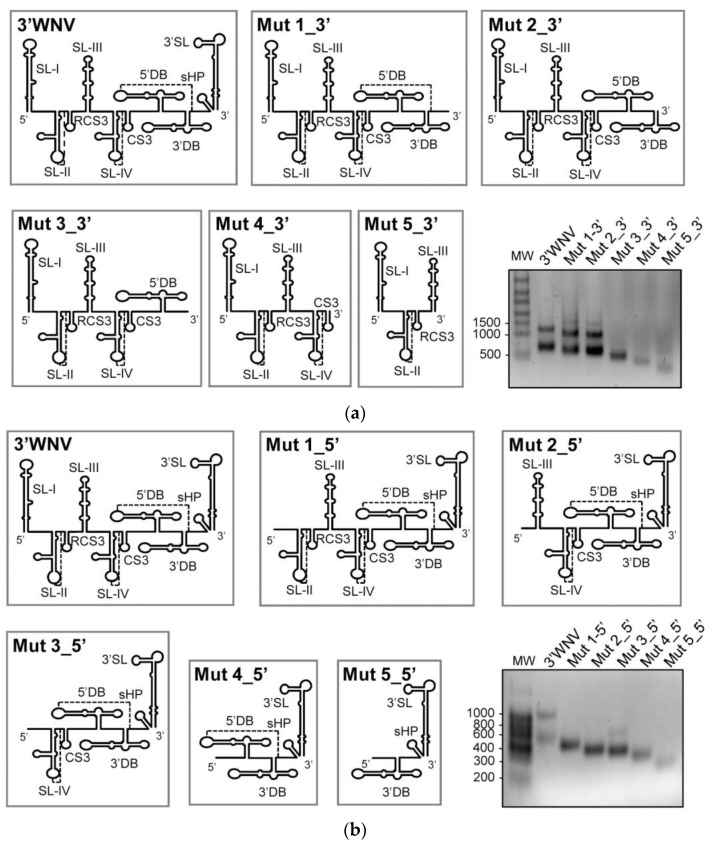
The elements SL-I and 3*′*DB are critical for dimer formation. Consecutive deletions of the independent structural elements within 3*′*WNV RNA were generated from the 3*′* (**a**) and 5*′* (**b**) ends to yield a collection of mutants that were assayed for their ability to dimerize, as described above. Different conformers were resolved as noted in Figure 2. Representative images of these assays are shown.

**Figure 5 ijms-24-05337-f005:**
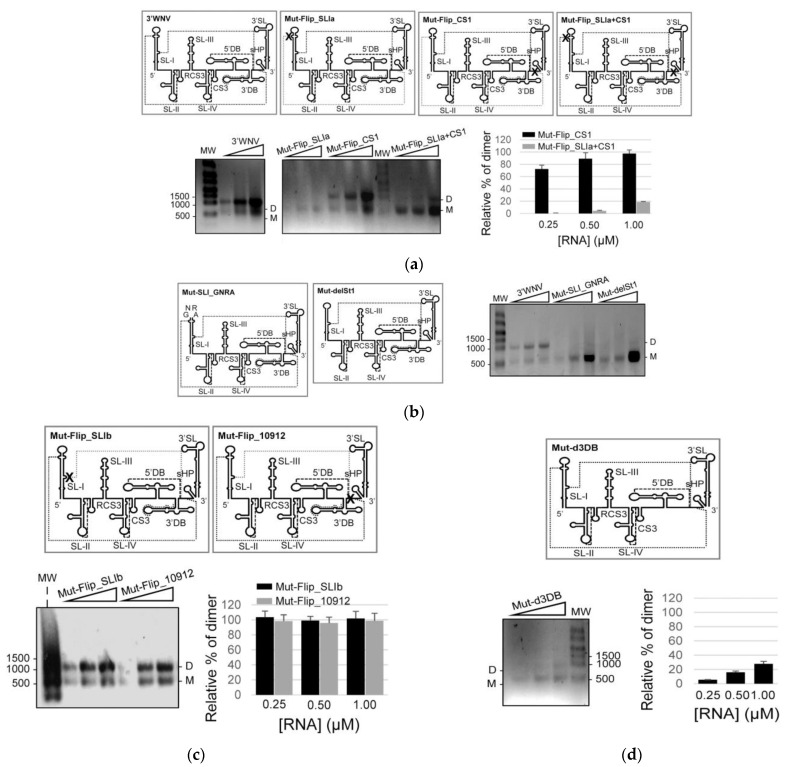
Specific sequences within the SL-I and the 3*′*DB elements are involved in the formation of dimeric conformers. A collection of mutants bearing different sequence variations in SL-I (**a**,**b**), the single-stranded region placed at the 3*′* end of the double-dumbbell elements (**c**) and the 3*′*DB (**d**) were generated and monitored for their capacity to form dimers. Conformers were resolved as noted in Figure 4. Histograms show the dimer rate for each construct relative to that shown by 3*′*WNV RNA. Data are the mean of three independent experiments ± standard deviation.

**Figure 6 ijms-24-05337-f006:**
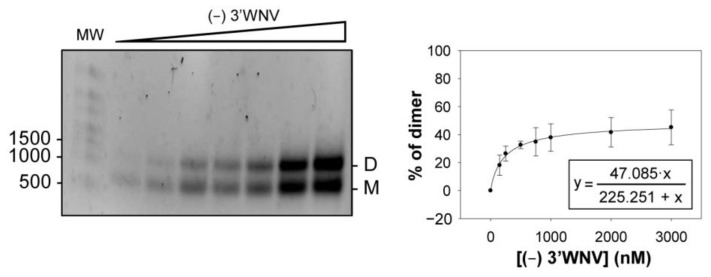
The (−) RNA of the WNV 3*′* UTR dimerizes in vitro. The antisense sequence of 3*′*WNV RNA ((−) 3*′*WNV) was synthesized and analyzed to evaluate its dimerization ability. Different concentrations of the (−) 3*′*WNV were incubated under dimerization reaction conditions, and dimers were resolved as described above (left panel). The percentage of dimers was quantified, and data were fitted to a non-linear equation to calculate the final reaction yield and the K_d_ values (right panel). Data are the mean of three independent assays ± standard deviation.

**Figure 7 ijms-24-05337-f007:**
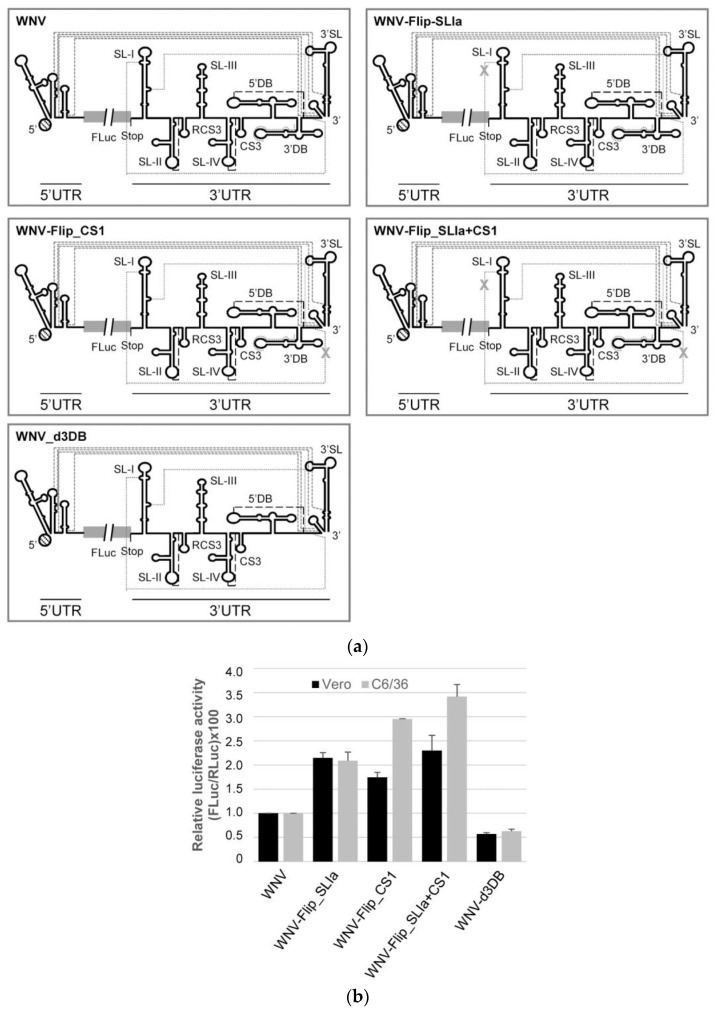
Changes in the dimerization efficiency correlate with WNV 5*′* UTR-dependent protein synthesis. (**a**) Diagrams show different constructs used in this study harboring the FLuc coding sequence flanked by the WNV 5*′* and 3*′* UTRs. The mutations described in Figure 5 were included in the 3*′* UTR to assay their role in translation regulation. (**b**) The translation efficiency of the constructs shown in (**a**) was measured in both Vero and C6/36 cell lines. Cells were cotransfected with each RNA construct and cap-RLuc RNA in order to normalize transfection efficiency. FLuc activity was measured at 4 h post-transfection. The histogram shows relative luciferase values relative to that obtained for the wt construct, WNV. Values are the mean of four independent assays ± standard deviation.

**Figure 8 ijms-24-05337-f008:**
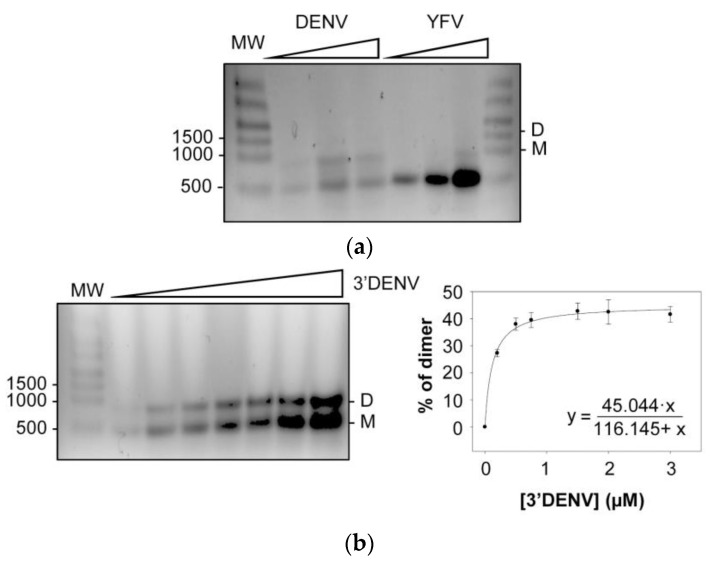
Dimerization of the genomic 3*′* UTR of DENV. (**a**) The genomic 3*′* UTRs of two flaviviruses that are closely related to WNV, DENV (3*′*DENV) and YFV (3*′*YFV), were subjected to dimerization assays. Conformers were resolved as noted in Figure 2. (**b**) Dimerization assay at increasing concentrations of 3*′*DENV RNA. Left panel: a representative image of these experiments. The percentage of dimers was quantified as described, and data were fitted to a non-linear equation (right panel). Values are the mean of three independent experiments ± standard deviation.

## Data Availability

The primer sequences used in this work will be provided upon request.

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
