# Peer review of "Inter- and Intramolecular RNA–RNA Interactions Modulate the Regulation of Translation Mediated by the 3′ UTR in West Nile Virus"

_ijms, 2023, doi:10.3390/ijms24065337_

Round 1

Reviewer 1 Report

This manuscript describes the inter- and intramolecular RNA-RNA interactions found in the West Nile virus (WNV) RNA genome. The authors proposed that the two structural elements, SL-I and 3’DB, are involved in the RNA-RNA interactions and regulate the translation efficiency. Formation of dimer was clearly demonstrated by the native agarose gel electrophoresis. And deletion mutants suggest the involvement of SL-I and 3’DB. However, the process of the dimerization might be complicated. In general, this manuscript will give important information on the structure and function of the viral RNA genome. To improve the clarity, please address the following points.

1. Is there any difference in dimerization efficiency between Mut-d3DB and Mut3-3’? Does the dimer bind appear for Mut3-3’ in higher concentration?

2. The authors stated that Mg2+ shows inverse correlation between the ion concentration and dimerization ability and Na+ had a positive effect on dimer formation. However, Mg2+ also shows a positive effect in the lower concentration region. Considering the higher effect of Mg2+ compare to Na+, the positive effect of Mg2+ in the lower concentration may correspond to the positive effect of Na+ in 10-100 mM range. Relating this, it is difficult to read the exact concentration of cations in Figure 2b and especially for 2c. The authors may include exact concentrations in the method section.

3. The region of the CS1 is not clear in Figure 3a. Please indicate the exact sequence of CS1.

4. This reviewer is curious for the gel experiment in which the Mut3_3’ and Mut2_5’ is mixed. If the intermolecular interaction occurs between SL-I and 3’DB, heterodimer can be observed. Please consider this.

Author Response

We thank the reviewer for the detailed review and helpful comments. We have addressed all the questions raised.

  1. Is there any difference in dimerization efficiency between Mut-d3DB and Mut3-3’? Does the dimer bind appear for Mut3-3’ in higher concentration?

We thank the reviewer for this interesting question. Indeed, the dimerization efficiency for Mut-d3DB is around 30%, while Mut3-3’ is unable to dimerize under the identical reaction conditions. Increasing the concentrations the result remains the same. The issue is that while Mut-d3DB lacks only domain 3’DB, Mut3-3’ has lost the 3’ end from the 5’ DB element. This could imply the acquisition of trapped conformations in the monomeric form or the existence of additional structural constraints placed at the sHP and 3’SL elements that would cooperate to form dimers. This last assumption is supported by the HMX results, which show structural determinants for dimerization placed at the 3’ CYC and the CS1 sequences, as well as in the apical loop of sHP (Figure 3c).

  1. The authors stated that Mg2+ shows inverse correlation between the ion concentration and dimerization ability and Na+ had a positive effect on dimer formation. However, Mg2+ also shows a positive effect in the lower concentration region. Considering the higher effect of Mg2+ compare to Na+, the positive effect of Mg2+ in the lower concentration may correspond to the positive effect of Na+ in 10-100 mM range. Relating this, it is difficult to read the exact concentration of cations in Figure 2b and especially for 2c. The authors may include exact concentrations in the method section.

We thank the reviewer for this helpful observation. We agree with the reviewer that high concentrations of Na+ could act by minimizing repulsive forces between sugar-phosphate backbones and thus favoring interactions like helical stacking. It is quite plausible that Mg2+ could exert a similar effect than Na+ at lower concentrations. We have slightly modified the sentence in order to fit the reviewer suggestion. In addition, the concentrations of Mg2+ and Na+ used in the assays have been added in the Materials and Methods section. 

  1. The region of the CS1 is not clear in Figure 3a. Please indicate the exact sequence of CS1.

We have modified the figure accordingly. 

  1. This reviewer is curious for the gel experiment in which the Mut3_3’ and Mut2_5’ is mixed. If the intermolecular interaction occurs between SL-I and 3’DB, heterodimer can be observed. Please consider this.

We thank the reviewer for this question. We have carried out an assay incubating Mut3_3’ and Mut2_5’, as suggested. In the case of the Mut3_3' construct, the acquisition of a dimeric conformation is observed, but at a very low rate (<1%), while with the 2_5' dimeric conformation is completely absent. Interestingly, they form a heterodimer but to a lesser extent than that detected when using the full-length 3’ UTR (see attached figure below). This is very interesting since it demonstrates the existence of an intermolecular interaction, likely involving SL-I and 3’DB, as suggested by the results presented in the manuscript. This assay also shows that additional structural elements of the 3’ UTR are required for efficient dimerization, as we have discussed in “comment 1”.

Reviewer 2 Report

The manuscript by Romero-Lopez, C., and collaborators describes how the 3´UTR of West Nile virus clearly describes a series of RNA elements that are responsible for inter- and intramolecular RNA-RNA interactions. However, at least based on how the experiments are described and the data are presented the only clear interactions are the intramolecular, but not the inter. Based on the following comments I suggest doing a major revision:

1.- Please clearly explain the data that shows how the mutant RNAs show intermolecular interactions. If these interactions are based on the HMX data, you need to better explain this data.

2.-The HMX methodology is very similar to the one obtained by SHAPE and other RNA-probing methods; however, the way the HMX data is shown in a way I have never seen. Normally, this kind of data is presented by different colors that indicate different ranges of reactivities. Why is figure 3 not showing such a range of interactions?

3.-Normally, when SHAPE is done the RNA is folded. Why was the RNA denatured for the HMX experiments? It is not clear what you can learn by modifying the RNA when is denatured. Please elaborate on why was performed like that.

4.-Finally, with respect to the HMX data. If we keep comparing it with a SHAPE-like analysis, it is not clear what is learned here. In SHAPE-like approaches, one measured the chemical reactivity of the RNA in the folded state and from that one can infer the secondary structure. However, given that the reaction is carried under denaturing conditions the reactivity of the NMIA reagent cannot be correlated to the RNA structure.

5.- The fitting of the % of dimer RNA as a function of RNA concentration is very concerning. What is the physical meaning of the amplitude of the reaction? Please cite a paper that demonstrates the validity of such a model. Please try other models like a 1:1 binding (Langmuir binding isotherm), n binding sites with non-cooperative interactions, and n binding sites with cooperative interactions (Hill´s equation). It is important to show which of these models best fits the data. I find the article by Comas-Garcia, M. et al eLife 2017 very instructive on how to carry out these three kinds of fits. This would apply to all the graphs where the data is fitted.

6.- Figure 2b. Why there are no experimental data for the graph? There is only a line. What model was used to fit this data? Why does the mobility of the RNA changes for the highest NaCl concentration? Also, the bands for the highest NaCl concentration are very dim; is the RNA denatured or is there a mistake in the amount of loaded RNA?

7.- Lines 141-142. The statement in that line contradicts the gel in figure 2c. As the amount of NaCl is increased there is a larger fraction of dimeric RNA.

8. Figure 3c and line 203. The NMIA reactivity is confusing for the proposed intra-RNA interactions between TL2 and nts 10,940. These nucleotides do not have the same reactivity, hence it is not clear what you are suggesting; the unpaired region in domain III has a high reactivity while the proposed interaction partner in TL-2 has no reactivity. Please explain this.

Also, what is the evidence for the interactions between SL-II and SL-IV with the respective regions that are indicated in the figure?

9.- Line 214-215 I disagree with this statement. Figure 3c proposes intra-molecular interactions; how does this data suggest its importance in dimerization? Please elaborate. It is clear that TL-2 has a palindromic sequence that suggests participating in dimerization, but this has not been shown at least at this point.  To show that the TL-2 participates in dimerization you will need to perform point mutations that disrupt the palindrome and test the % of dimerization.

10.- Figure 4a and b is very confusing because the mutants go from being observed as well-defined bands to smears. This would suggest that either mutations 4_3' and 5_3' completely unfold the RNA, the RNAs are degraded, or the gel cannot resolve these RNAs because they are too short. Also, why does the intensity of bands of the mutant RNA decrease compared to the other RNAs?

11.- Line 234. I disagree with this statement, you can´t see a dimer but only a smear.

12.- Figure 5. Please add a figure legend for the graphs that have black and gray bars.

13.- Figure 5. The data for the Mu-Flip-SL1 is very concerning as no bands are observed. This suggests that either there is degradation, or the amount of RNA is lower than expected. You can´t conclude anything from this RNA because the RNA is barely visible. Also, the amount of Mut-d3DB is barely visible in the gel. please repeat these gels.

14.- Line 311 and Figure 6. What is the Kd? Also, please take into consideration comment 5 for this graph and the other ones that involve fitting.

15.- Section 2.6. Please explain how the identity of the 3´UTR and its ability to dimerize affect the expression of FLuc. This might be obvious to the authors but the logic behind the experiment must be clear to all readers.

16.- Figure 8. Why there are no error bars in the graph?

17.- Line 398. What do you mean by RNA assembly?

18.- Lines 414-415. That is not true, there are several physical models that can fit such curves and if you want to make a such a claim you need to show that other accepted models cannot fit the data. In other words, to claim that there are two affinity sites you need to show that the other binding models do not properly fit the data. As mentioned before in Comas-Garcia, M. eLife 2017 there is a good analysis of how this has to be done.

19.- Line 428.  What do you mean by "dynamic folding"?

Author Response

We thank the reviewer for the detailed review and all helpful suggestions. We have addressed all the questions raised. Before going into the point-by-point response to the comments we must say that we do not understand the reviewer´s statement that “at least based on how the experiments are described and the data are presented the only clear interactions are the intramolecular, but not the inter”. A priori we cannot agree with such statement. As the experiments show, two conformers are observed by native agarose gel electrophoresis, whose sizes correspond to those of the monomer and dimer conformations. Dimer formation undoubtedly implies the establishment of an interaction of two molecules, therefore intermolecular interactions.

We really appreciate the comments 1-4 questioning the design of the HMX experiments and, therefore, the conclusions drawn from them. We consider that the reviewer is most probably unfamiliar with the technique. Therefore, we consider it is important to explain the following first:

SHAPE analysis and HMX are two different techniques that only have in common the chemistry used. They provide very different information that could be complementary. SHAPE is used to know the structure of an RNA molecule. Therefore, chemical modification is performed on a folded RNA molecule, as indicated by the reviewer. While HMX provides information about the nucleotides required for the acquisition of a specific folding or conformation. It provides information on the required structural features at precise nucleotides that are necessary for a specific interaction to occur, either because they are directly involved in that interaction or because they participate in the formation of the competent structure for this interaction. Therefore, chemical modification is performed under denaturing conditions. The modification consists in the introduction of a bulky adduct under controlled conditions to achieve one modified nucleotide per molecule (or less). Then RNA molecules are allowed to fold and to interact with a different molecule e.g. a protein or an RNA, as in this case. A subset of the nucleotide modifications will interfere with the required folding or interaction. Then, the different conformers or complexes are partitioned and analysed. Comparison of nucleotide reactivity between the two molecular conformations, or between bound and unbound molecules, will provide information about the nucleotides involved in the formation of a particular conformation or binding.

1.- Please clearly explain the data that shows how the mutant RNAs show intermolecular interactions. If these interactions are based on the HMX data, you need to better explain this data.

RNAStructure analysis predicts intramolecular contacts. However, under RNA crowded conditions, such as those chosen for the dimerization assays, these interactions can be established in intermolecularly (by strand exchange, kissing-loop and nucleotide complementarity). RNA dimerization is a common strategy to achieve functional RNAs. For instance, many unmodified tRNAs exhibit a strong tendency to dimerize and accumulate in RNP granules as a translational machinery store.

2.-The HMX methodology is very similar to the one obtained by SHAPE and other RNA-probing methods; however, the way the HMX data is shown in a way I have never seen. Normally, this kind of data is presented by different colors that indicate different ranges of reactivities. Why is figure 3 not showing such a range of interactions?

The data obtained by HMX do not provide the same information than those obtained by SHAPE (see response to comment 3). While the HMX technique uses the same chemistry than SHAPE, the methodology itself is different, as noted by the reviewer. Therefore, data obtained by both techniques cannot be represented in the same way. In the case of HMX, the relevant information concerns to significant changes in reactivity between different conformations, but the reactivity data for a specific conformer itself is not so critical, as long as this reactivity is higher than 0.3 (as the reviewer will know, this is a widely accepted threshold value to assume positive reactivity). The application of statistics to calculate the correlation score, as we have done, renders those positions that show significant variations in their reactivity between the analyzed conformations. This is what we have represented in figures 3b and 3c. We have relied on the results reported by Dr. K. Weeks and cols. (Homan et al., 2014, Biochemistry, 53:6825-6833, reference 22 of the manuscript), who are one of the most relevant groups in the development of SHAPE and SHAPE-related methodologies. They described this procedure based on the calculation of correlation scores and used similar representations to those shown in the manuscript. We have also previously used thesecalculations to analyze the nucleotides required for the dimerization of HCV genome (Romero-López et al., 2017, Sci Rep, 7, 43415). A sentence citing this fact has been included in the Materials and Methods section.

3.-Normally, when SHAPE is done the RNA is folded. Why was the RNA denatured for the HMX experiments? It is not clear what you can learn by modifying the RNA when is denatured. Please elaborate on why was performed like that.

4.-Finally, with respect to the HMX data. If we keep comparing it with a SHAPE-like analysis, it is not clear what is learned here. In SHAPE-like approaches, one measured the chemical reactivity of the RNA in the folded state and from that one can infer the secondary structure. However, given that the reaction is carried under denaturing conditions the reactivity of the NMIA reagent cannot be correlated to the RNA structure.

Backbone dynamics can be monitored by SHAPE with the use of chemical reagents specific for the 2′-OH group, which nucleophilicity is dependent on the flexibility of the modified nucleotide. The SHAPE reagents introduce bulky 2’-O-adducts that can affect the final folding of the RNA. Therefore, it is critical to perform these modifications under those conditions that achieve evenly distributed modifications. Further, it is noteworthy that reactivities are modulated by both secondary and tertiary interactions, and it is usually difficult to infer the relative influence of each type of interaction.

For this reason, additional methods have been developed to map RNA secondary and tertiary interactions. These methods are based on the modification of an RNA, either with chemical probes (the so-called modification interference) or by incorporating nucleotide substitutions. In both cases, a disruption in the native structure must be generated by the incorporation of modifications. In modification interference (Conway et al., 1989, Methods Enzymol, 180:369-379; Clarke, 1999, Methods Mol Biol, 118:73- 91), which is the general technique on which HMX (2′-hydroxyl molecular interference) is based, an RNA is treated to introduce chemical modifications, usually at the nucleobases. The introduction of chemical modifications as bulky adducts must be controlled to occur as an evenly event. Further, the RNA is modified under denaturing conditions, to introduce modifications independently of the flexibility of each nucleotide. Therefore, all nucleotides will be modified with the same probability. A subset of these modifications will interfere with the RNA folding. Then, the RNA is subjected to a partitioning experiment to distinguish functional (in our particular case, dimers) from non-functional molecules (monomers). By partitioning the sample into dimers and monomers states, nucleotides whose modification disrupts dimerization are identified.

HMX assays use the classical molecular interference methodology with SHAPE chemistry (Homan et al., 2014, Biochemistry, 53:6825-6833). In these analyses, NMIA is chosen as a modifying agent since it works well at high temperatures by introducing bulky adducts that interfere with RNA folding in particularly crowded regions. This technique is selective for characterizing essential nucleotides groups and for establishing hierarchical relationships between residues required for the acquisition of the 3D folding (Homan et al., 2014, Biochemistry, 53:6825-6833). For all these reasons, HMX offers completely different information to that recovered by simple SHAPE assays. Both of them are equally useful methods, and in fact they can be combined to get a more complete structural map. In our particular case, applying the SHAPE methodology would render a noisy reactivity pattern since dimerization is not complete, therefore the mix of monomers and dimers would react with NMIA at similar extent, hindering the deconvolution of each conformer contribution.

 5.- The fitting of the % of dimer RNA as a function of RNA concentration is very concerning. What is the physical meaning of the amplitude of the reaction? Please cite a paper that demonstrates the validity of such a model. Please try other models like a 1:1 binding (Langmuir binding isotherm), n binding sites with non-cooperative interactions, and n binding sites with cooperative interactions (Hill´s equation). It is important to show which of these models best fits the data. I find the article by Comas-Garcia, M. et al eLife 2017 very instructive on how to carry out these three kinds of fits. This would apply to all the graphs where the data is fitted.

We thank the reviewer for this interesting observation. However, we have developed a biochemical approach, which provides data with different meaning than those obtained by molecular dynamics or other physical methods, as suggested by the reviewer. In fact, we obtain affinity data (Kd, a semiquantitative parameter) (Jarmoskaite et al., 2020, eLife 2020;9:e57264), but not thermodynamic or kinetic results, as those derived from molecular dynamics. We tested different fits with the SigmaPlot software and the one chosen was the best fitting curve, with an adjust R2 value of 0.9955. In addition, cooperative interactions were not predicted since the Hill coefficient was ~ 1.  

6.- Figure 2b. Why there are no experimental data for the graph? There is only a line.

We apologize for the error, the data were inadvertently removed when constructing the figure. Data have been added to Figure 2c.

What model was used to fit this data?

These data are just a scattered-line plot, they do not fit to any equation since the aim of this assay is just a qualitative observation of the dimer behavior in the presence of different ionic conditions. The same applies to the experiments at different Mg2+ concentrations.

Why does the mobility of the RNA changes for the highest NaCl concentration?

It is an artifact of the electrophoresis due to the high concentration of Na+. It induces a retard in the mobility of the RNA.

Also, the bands for the highest NaCl concentration are very dim; is the RNA denatured or is there a mistake in the amount of loaded RNA?

As we have mentioned in the previous comment, this is an artifact of the electrophoresis. It is a common effect seen in agarose or acrylamide electrophoresis when the sample conditions change or under high ionic strength. The resolution of a specific sample changes when modifying the composition of the sample buffer. In this particular case, clearly, the increase in [Na+] alters the resolution of the samples.

Regarding the apparently lower amount of RNA in the samples with higher [Na+] we attribute this effect to the high concentration of positive ions (Na+), which may hinder the intercalation of the fluorescent agent that is used for the visualization of the RNA (RedSafeTM, see Materials and Methods section).

7.- Lines 141-142. The statement in that line contradicts the gel in figure 2c. As the amount of NaCl is increased there is a larger fraction of dimeric RNA.

We agree with the reviewer on this observation, but it must be noted that this sentence does not refer to our results in particular, but is a general observation for all RNAs. A reference describing this fact has been included to clarify this point (Fischer et al., 2018, Nucleic Acids Research, 46:4872-4882).

8.- Figure 3c and line 203. The NMIA reactivity is confusing for the proposed intra-RNA interactions between TL2 and nts 10,940. These nucleotides do not have the same reactivity, hence it is not clear what you are suggesting; the unpaired region in domain III has a high reactivity while the proposed interaction partner in TL-2 has no reactivity. Please explain this. Also, what is the evidence for the interactions between SL-II and SL-IV with the respective regions that are indicated in the figure?

What the referee ask about SLII and SLIV interactions is not derived from our data, but it is a previously well demonstrated contact (Sztuba-Solinska et al., 2013, Nucleic Acids Res. 2013, 41:5075–5089; Villordo et al., 2015, PLoS Pathog. 11:1–22; Villordo et al., 2016, Trends Microbiol. 24:270–283). It must be mentioned, in addition, that reactivity data are obtained from HMX assays, which renders different information that SHAPE, as argued previously. What we are representing in the figure are differences in reactivity between the two conformers.

9.- Line 214-215 I disagree with this statement. Figure 3c proposes intra-molecular interactions; how does this data suggest its importance in dimerization? Please elaborate.

We agree with the reviewer that we have no HMX data for SL-I. The participation of SL-I is inferred from the structural model prediction. Therefore, this statement has been modified in the manuscript.

Applying HMX, we have identified several structural determinants that are key for the acquisition of the dimer. Please, consider the HMX information provided in previous comments. In addition, it must be noted that intramolecular contacts predicted by RNAStructure (and modeled from the NMIA reactivity values) can be displaced towards intermolecular interactions in RNA crowded environments. Therefore, it is perfectly assumed that the detected dimers at high RNA concentrations are formed by the same interactions that account in an intramolecular way at low RNA concentrations.

It is clear that TL-2 has a palindromic sequence that suggests participating in dimerization, but this has not been shown at least at this point.  To show that the TL-2 participates in dimerization you will need to perform point mutations that disrupt the palindrome and test the % of dimerization.

We thank the reviewer for this interesting suggestion, in fact, it is something we have in mind to perform soon. However, a detailed analysis of this requires the generation of a significant number of single mutants and compensatory mutations that requires a lot of work. This is something we intend to do. On the other hand, the introduction of mutations in TL2 or in the closing stem would greatly disturb the folding of this region, which according to HMX data must be folded in a precise way to achieve dimerization, so it is possible that interpretation of the results would not be easy.

10.- Figure 4a and b is very confusing because the mutants go from being observed as well-defined bands to smears. This would suggest that either mutations 4_3' and 5_3' completely unfold the RNA, the RNAs are degraded, or the gel cannot resolve these RNAs because they are too short.

We appreciate the observation of the reviewer. The Mut4-3’ and Mut5_3’ constructs might fold into different suboptimal conformations that could not be resolved under our conditions, as proposed by the secondary structure modelling performed with RNAStructure. These suboptimal conformations would be visualized as smears. Since the smear shows a lower mobility than the major conformer, degradation of the RNA is discarded. 

Also, why does the intensity of bands of the mutant RNA decrease compared to the other RNAs?

Mutants bear sequential deletions, therefore a less amount of RedSafeTM can be intercalated into the RNA. This is the reason of their lower intensity.

11.- Line 234. disagree with this statement, you can´t see a dimer but only a smear.

We regret to disagree with the reviewer on this point. Mut3_5’ does not form a smear. Mut5_3’ does it, but a discrete band of the predicted dimer size can be also observed. The smear is the consequence of the suboptimal folding of a small population of molecules. 

12.- Figure 5. Please add a figure legend for the graphs that have black and gray bars.

 We thank the reviewer for this helpful comment. We have added the legend accordingly.

13.- Figure 5. The data for the Mu-Flip-SL1 is very concerning as no bands are observed. This suggests that either there is degradation, or the amount of RNA is lower than expected. You can´t conclude anything from this RNA because the RNA is barely visible. Also, the amount of Mut-d3DB is barely visible in the gel. please repeat these gels.

We acknowledge the reviewer for this observation. We have added below two images of the same experiment to clarify this point. We decided to include the image shown in Figure 5 since it contains all the mutant constructs in the same gel, but, as it has been mentioned, it is only a representative image from three independent experiments. 

14.- Line 311 and Figure 6. What is the Kd? Also, please take into consideration comment 5 for this graph and the other ones that involve fitting.

As we have argued in comment 5, Kd is a semiquantitative parameter that reflects the affinity of a molecule for its target. As it can be deduced from the equation, it is the concentration of RNA required to reach the half of the maximum yield of the reaction (the so-called amplitude or Bmax, see Materials and Methods section). This means that the higher Kd, the lower affinity, since more molecules are required to achieve the half of the maximum reaction yield. This parameter has been largely used to calculate the affinity of RNA molecules for their targets and it is well validated (Jarmoskaite et al., 2020, eLife 2020;9:e57264). Since we have used a biochemical strategy, it is difficult to calculate robust thermodynamic or kinetic parameters, as is done with other physical methods that are beyond the scope of the present manuscript.

15.- Section 2.6. Please explain how the identity of the 3´UTR and its ability to dimerize affect the expression of FLuc. This might be obvious to the authors but the logic behind the experiment must be clear to all readers.

The results show that sequences that affect dimerization also affect translation, although in different ways. For example, modifications in SL-I produce an increase in translation, possibly due to a deficit in Msi protein recruitment, as argued in the discussion section (Line 454). This is inversely correlated with the increase in dimerization observed for these constructs. However, the absence of the 3’DB element promotes a decrease in both translation and dimerization efficiencies. These results support that the control of WNV translation mediated by the 3’ UTR is a multifactorial phenomenon that depends on the establishment of a proper balance between different elements, as previously described (Berzal-Herranz et al., 2023, IJMS, 23). The molecular mechanisms that control the regulation of translation mediated by the structural RNA elements of the 3’ UTR of WNV genome, but also for any other RNA virus, are currently unknown. Although it is a hot topic it is not obvious and unfortunately, we cannot give a logical explanation. We hope that our work can contribute to deciphering these mechanisms.

16.- Figure 8. Why there are no error bars in the graph?

We are sorry for this mistake. We have added the corresponding error bars.

17.- Line 398. What do you mean by RNA assembly?

RNA assembly includes RNA folding but also considers quaternary interactions, as the formation of dimers. 

18.- Lines 414-415. That is not true, there are several physical models that can fit such curves and if you want to make a such a claim you need to show that other accepted models cannot fit the data. In other words, to claim that there are two affinity sites you need to show that the other binding models do not properly fit the data. As mentioned before in Comas-Garcia, M. eLife 2017 there is a good analysis of how this has to be done.

We have modified this sentence according to reviewer’s suggestion. 

19.- Line 428.  What do you mean by "dynamic folding"?

It means that the discrete structural units switch from one state to another, promoting the breathing of the structure and the functionality of the molecule. It is a widely accepted concept.

Round 2

Reviewer 2 Report

I want to thank the authors for the answers, and I want to apologize for a mistake a made. I meant that “at least based on how the experiments are described and the data are presented the only clear interactions are the intremolecular, but not the intra”. I inverted the words; most of the data comes from gel retardation assays that are sensitive when an RNA multimerizes; which is normally product of intermolecular interactions. I did not find compelling evidence to say with great confidence that is clear data about intramolecular interactions.

With respect to the confusions between HMX and SHAPE I raised these points because only and expert on HMX will immediately understand the differences. However, SHAPE-like methodologies are far more common than HMX so the authors should not expect to everyone to immediately know the key points that make the data from these two techniques learn different things. Therefore, I think that it is important to briefly summarize the through explanation they gave me into the discussion. This would greatly help any reader to fully appreciate their work.

I disagree with their answer for question no. 5. Those gels show the differences in populations between a monomer and a dimer. First of all, that is not a biochemical approach, it is physical method based on the electrophoretic mobility of two or more macromolecules in a porous media. Second of all, if you fit any data the formula must have a physical meaning, especially when you mention in the text that you used that formula to calculate a Kd. The equation used in those fittings cannot be used to calculate a Kd unless the physical meaning of the parameters is determined. Therefore, it is not acceptable to say that your function has an amplitude and you do not justify this parameter. You cannot introduce “an amplitude” without a proper justification. What you did what just to use a polynomial solution to fit the data. Based on this equation there is very little or nothing that can be said about a Kd.

The statement “In fact, we obtain affinity data (Kd, a semiquantitative parameter) (Jarmoskaite et al., 2020, eLife 2020;9:e57264), but not thermodynamic or kinetic results, as those derived from molecular dynamics” shows that the authors do not understand the concept of Kd. A Kd is by definition a THERMODYNAMIC parameter. A Kd is nothing more that the exponent of the (minus) Gibbs free energy of a binding process divided by the thermal energy (RT). Since it is derived from the Gibbs free energy it is a thermodynamic parameter. Furthermore, their comment about molecular dynamics makes no sense. There is no need to use molecular dynamics to understand a monomer-dimer interactions and to estimate the dissociation constant of this process. This has been study for more than 100 years and one can easily calculate the dissociation constant without having to use molecular dynamics. Nonetheless, if the authors still want to say anything about the Kd they must compare the fitting of a Langmuir isotherm, a two-binding-site isotherm to that of their equation that contains that “amplitude” term. Also, I should mention that just because R^2 is close to 1 it does not mean that equation as a physical meaning. Based on these arguments, I would strongly suggest to the authors and editor to do one of two things: i) either remove any comment about Kd, or 2) to compare the fitting of their equation to that of known and well-stablished equations used to measure association and justify the physical meaning of the amplitude term,

            The explanation that the authors give about why at high salt concentration the bands took less intense that at lower NaCl concentrations. If this were to be the case, then the overall intensity of the bands would decrease as a function of salt concentration. However, in their data the difference between the highest and the second-highest salt concentrations is abrupt.

            With regards to comment 7 this is confusing. Your data shows one thing, but your argument is the opposite. I understand now you are referring to someone else´s data. Nonetheless, what is your explanation of why you observe the opposition to Fisher et al 2018?

            In answer to point 8, if their argument is not based on their data then they should include the arguments that use the data published by Joanna Sztuba-Solinska and other groups. Having this information in the article is extremely important.

            I understand the comments about mutating the palindromic sequence of TL-2. I think that it would help the manuscript to discuss this limitation in the discussion.

            With respect to question 10. If the mutant RNAs are not folding properly, then it is not clear what can learn about dimerization. Figure 4 says “The elements SL-I and 3’DB are critical for dimer formation”, but if the RNA is unfolding because of the mutations then the only thing you can conclude is that those mutations affect the overall stability of the RNA to fold into the proper structure. This by itself is interesting. If the RNA is being unfolded, then it would make sense that the RedSafe dye would bind with lower affinity.

            For question 11, I am sorry, I should have specified that the smear I was referring was for mutant 5_3. Given the amount of smear for this mutant you cannot really conclude anything about the ability of this RNA to dimerize; there might be not only dimers but perhaps other high-order structures.

            I disagree with the answer to point 14. A Kd is not a semiquantitative parameter, it is by definition a quantitative parameter. I aware of the Jarmoskaite paper, and one of the problems with this paper is that arbitrary amplitude, this is why I asked to compare the fitting between this equation and that of a Langmuir isotherm which is the most common equation used to measure 1:1 binding that does not introduce arbitrary fitting parameters such as A. I also reiterate my comment that a gel retardation assay is not a biochemical assay but a physical one and that the argument about kinetics (based on my observation makes no sense). There are plenty of examples where a dissociation constant can be extracted from an gel retardation assay: 10.1093/nar/gkh754, 10.1006/jmbi.1995.0278, Carey, J. PNAS 1988, 10.1074/jbc.M007876200, 10.1128/mr.56.4.509-528.19. In fact the last paper from Lane at al clearly explains why you can measure Kd in gel retardation asays.

            I think that the authors did not get my point made in comment 15. I am asking to explain how changing the structure of the 3´UTR affect the expression of the FLuc not for my own sake but for the reader to understand the experiment. I would kindly ask the authors to include this explanation in the manuscript so readers that are not expert on WNV can understand how modifying the 3`UTR affect the ability of the mRNA to be translated.

Author Response

We thank the reviewer for this illustrative discussion. We have answered the questions raised and modified the manuscript accordingly.

I want to thank the authors for the answers, and I want to apologize for a mistake a made. I meant that “at least based on how the experiments are described and the data are presented the only clear interactions are the intremolecular, but not the intra”. I inverted the words; most of the data comes from gel retardation assays that are sensitive when an RNA multimerizes; which is normally product of intermolecular interactions. I did not find compelling evidence to say with great confidence that is clear data about intramolecular interactions.

We thank the reviewer for clarifying this point. With our current data, we cannot exclude the existence of such intramolecular interactions at low RNA concentrations. We have modified the sentences in which this idea is included.

With respect to the confusions between HMX and SHAPE I raised these points because only and expert on HMX will immediately understand the differences. However, SHAPE-like methodologies are far more common than HMX so the authors should not expect to everyone to immediately know the key points that make the data from these two techniques learn different things. Therefore, I think that it is important to briefly summarize the through explanation they gave me into the discussion. This would greatly help any reader to fully appreciate their work.

We appreciate this comment, we have included a brief description of HMX in the corresponding results section (lines 151-156).

Based on these arguments, I would strongly suggest to the authors and editor to do one of two things: i) either remove any comment about Kd, or 2) to compare the fitting of their equation to that of known and well-stablished equations used to measure association and justify the physical meaning of the amplitude term,

We want to acknowledge the reviewer for the illustrative explanation about Kd significance. We are aware that we have used a simplification of the equation. We have removed all the comments concerning to Kd and the conclusions we had derived (lines 19, 121-122, 315, 390-392 and 413-415. We have also added a modification in the Materials and Methods section in order to avoid misinterpretations (lines 564-568).

The explanation that the authors give about why at high salt concentration the bands took less intense that at lower NaCl concentrations. If this were to be the case, then the overall intensity of the bands would decrease as a function of salt concentration. However, in their data the difference between the highest and the second-highest salt concentrations is abrupt.

We appreciate this comment, we have quantified the total intensity of the bands in the referred lanes:

Monomer

Dimer

Lane 6

40959252

43337511

Lane 7

29716122

38649660

Although there are differences in total intensity, these are not as abrupt as might be inferred from the image. It should be noted that due to the presence of Na+ loading the RedSafeTM intercalation may be altered and result in a lower intensity. This would not be a linear process and would justify these differences.      

With regards to comment 7 this is confusing. Your data shows one thing, but your argument is the opposite. I understand now you are referring to someone else´s data. Nonetheless, what is your explanation of why you observe the opposition to Fisher et al 2018?

We do not agree with the statement that our data and Fischer's data are opposite. We propose that Mg2+ stabilizes certain conformations at high concentrations that are incompetent for dimerization. This hypothesis is in good concordance with Fischer’s findings, who argue that Mg2+ preferentially binds and strongly stabilize helical regions, while Na+ would stabilize other complex foldings. In addition, Fischer et al also observed that “Both Na+ and Mg2+ ions bind sequence specific and also to specific binding sites. In both helical and complex folded structures certain RNA atoms are preferred. In complex folded structures atoms are available for binding that are not sterically accessible for ions in helical RNA structures” Finally, “At low ion concentrations a larger fraction of the Na+ and Mg2+ ions are in direct contacts with the RNA in our simulations than at higher concentrations. When, however, both Na+ and Mg2+ ions are present, more Mg2+ ions are closer to the RNA (distance less than 10 Å) than Na+. This is in agreement with the ‘ion atmosphere’ as described by Lipfert et al. (37). It seems therefore that the overall salt concentration should be factored in when considering the properties of the ‘ion atmosphere’.” This statement would be related with the fact that the system can be saturated by high Mg2+ concentrations, leading to a structurally trapped state, which would prevent dimerization. It must be noted that helical stabilization by Mg2+ has been also observed by other authors (Garcia-Sacristan et al., 2015, Nucleic Acids Research, 43:565–580, https://doi.org/10.1093/nar/gku1299)

All these arguments would support our findings and they are not opposite to the previously reported data.

In answer to point 8, if their argument is not based on their data then they should include the arguments that use the data published by Joanna Sztuba-Solinska and other groups. Having this information in the article is extremely important.

This reference is already included in the Introduction section, in which we mention the importance of these PK elements (line 74). In point 8, reviewer argues for differences in NMIA reactivity in TL2 and the unpaired region in domain III; however, our NMIA reactivity data (Figures 3a and 3b) are similar in the monomeric conformer for both regions, but significantly different in TL2 between monomer and dimer. This is what we use to argument the idea that TL2 must be “unflexible” (likely “paired”) in the dimer. This argument is exclusively based on our HMX data. This does not exclude the observation of Sztuba-Solinska and previous reports, since they used different RNA concentrations in their assays.

I understand the comments about mutating the palindromic sequence of TL-2. I think that it would help the manuscript to discuss this limitation in the discussion.

A sentence discussing this observation has been included in lines 424-426.

With respect to question 10. If the mutant RNAs are not folding properly, then it is not clear what can learn about dimerization. Figure 4 says “The elements SL-I and 3’DB are critical for dimer formation”, but if the RNA is unfolding because of the mutations then the only thing you can conclude is that those mutations affect the overall stability of the RNA to fold into the proper structure. This by itself is interesting. If the RNA is being unfolded, then it would make sense that the RedSafe dye would bind with lower affinity.

We appreciate this interesting observation. However, we have not said that the mutants are not properly folded or unfolded, we have said that they “might fold into different suboptimal conformations that could not be resolved under our conditions, as proposed by the secondary structure modelling performed with RNAStructure. These suboptimal conformations would be visualized as smears.“ It is noteworthy that these suboptimal foldings (smears) are just a very little fraction of the total RNA and therefore, their presence should not disrupt the global structural balance. The lower band intensity is related with the amount of RedSafe that is intercalated, since mutants are smaller than wt, a lesser amount of RedSafe can be bound to these RNAs and, therefore, the band intensity diminishes in a correlative manner.

For question 11, I am sorry, I should have specified that the smear I was referring was for mutant 5_3. Given the amount of smear for this mutant you cannot really conclude anything about the ability of this RNA to dimerize; there might be not only dimers but perhaps other high-order structures.

We thank for this clarification, we have added a short explanation in lines 244-246.

I disagree with the answer to point 14. A Kd is not a semiquantitative parameter, it is by definition a quantitative parameter. I aware of the Jarmoskaite paper, and one of the problems with this paper is that arbitrary amplitude, this is why I asked to compare the fitting between this equation and that of a Langmuir isotherm which is the most common equation used to measure 1:1 binding that does not introduce arbitrary fitting parameters such as A. I also reiterate my comment that a gel retardation assay is not a biochemical assay but a physical one and that the argument about kinetics (based on my observation makes no sense). There are plenty of examples where a dissociation constant can be extracted from an gel retardation assay: 10.1093/nar/gkh75410.1006/jmbi.1995.0278, Carey, J. PNAS 1988, 10.1074/jbc.M00787620010.1128/mr.56.4.509-528.19. In fact the last paper from Lane at al clearly explains why you can measure Kd in gel retardation asays.

Thank you again for this illustrative explanation, as we have mentioned before, we have removed all the comments referred to Kd in the manuscript.

I think that the authors did not get my point made in comment 15. I am asking to explain how changing the structure of the 3´UTR affect the expression of the FLuc not for my own sake but for the reader to understand the experiment. I would kindly ask the authors to include this explanation in the manuscript so readers that are not expert on WNV can understand how modifying the 3`UTR affect the ability of the mRNA to be translated.

Thank you for this clarification. We have added a short paragraph explaining this point in lines 322-323 and  461-464.
